# Serum Vitamin D Affected Type 2 Diabetes though Altering Lipid Profile and Modified the Effects of Testosterone on Diabetes Status

**DOI:** 10.3390/nu13010090

**Published:** 2020-12-30

**Authors:** Lulu Wang, Xue Liu, Jian Hou, Dandan Wei, Pengling Liu, Keliang Fan, Li Zhang, Luting Nie, Xing Li, Wenqian Huo, Tao Jing, Wenjie Li, Chongjian Wang, Zhenxing Mao

**Affiliations:** 1Department of Epidemiology and Biostatistics, College of Public Health, Zhengzhou University, Zhengzhou 450001, Henan, China; 202012272024844@gs.zzu.edu.cn (L.W.); liuxue_sdu@mail.sdu.edu.cn (X.L.); houjian1988@zzu.edu.cn (J.H.); weidandan@gs.zzu.edu.cn (D.W.); zdlpl1234@gs.zzu.edu.cn (P.L.); Ksssss@gs.zzu.edu.cn (K.F.); zhangli_bonnie@163.com (L.Z.); tjwcj2008@zzu.edu.cn (C.W.); 2Department of Occupational and Environmental Health Sciences, College of Public Health, Zhengzhou University, Zhengzhou 450001, Henan, China; nlt@gs.zzu.edu.cn (L.N.); huowenqian@zzu.edu.cn (W.H.); 3Department of Nutrition and Food Hygiene, College of Public Health, Zhengzhou University, Zhengzhou 450001, Henan, China; lixing530@zzu.edu.cn (X.L.); Lwj@zzu.edu.cn (W.L.); 4School of Public Health, Tongji Medical College, Huazhong University of Science and Technology, Wuhan 430074, Hubei, China; jingtao@hust.edu.cn

**Keywords:** vitamin D, testosterone, type 2 diabetes mellitus, impaired fasting glucose, lipid profile, interactive effects

## Abstract

Numerous studies have investigated the associations between serum vitamin D or testosterone and diabetes; however, inconsistencies are observed. Whether there is an interaction between vitamin D and testosterone and whether the lipid profile (total cholesterol (TC), triglyceride (TG), high-density lipoprotein cholesterol (HDL-C), and low-density lipoprotein cholesterol (LDL-C)) mediates the association between vitamin D and diabetes is unclear. To investigate the effect of vitamin D and testosterone on impaired fasting glucose (IFG) or type 2 diabetes mellitus (T2DM), 2659 participants from the Henan Rural Cohort were included in the case-control study. Generalized linear models were utilized to estimate associations of vitamin D with IFG or T2DM and interactive effects of vitamin D and testosterone on IFG or T2DM. Principal component analysis (PCA) and mediation analysis were used to estimate whether the lipid profile mediated the association of vitamin D with IFG or T2DM. Serum 25(OH)D_3_, 25(OH)D_2_, and total 25(OH)D levels were negatively correlated with IFG (odds ratios (*OR*s) (95% confidence intervals (*CI*s)): 0.99 (0.97, 1.00), 0.85 (0.82, 0.88), and 0.97 (0.96, 0.98), respectively). Similarity results for associations between serum 25(OH)D_2_ and total 25(OH)D with T2DM (*OR*s (95%*CI*s): 0.84 (0.81, 0.88) and 0.97 (0.96, 0.99)) were observed, whereas serum 25(OH)D_3_ was negatively correlated to T2DM only in the quartile 2 (Q2) and Q3 groups (both *p* < 0.05). The lipid profile, mainly TC and TG, partly mediated the relationship between 25(OH)D_2_ or total 25(OH)D and IFG or T2DM and the proportion explained was from 2.74 to 17.46%. Furthermore, interactive effects of serum 25(OH)D_2_, total 25(OH)D, and testosterone on T2DM were observed in females (both *p* for interactive <0.05), implying that the positive association between serum testosterone and T2DM was vanished when 25(OH)D_2_ was higher than 10.04 ng/mL or total 25(OH)D was higher than 40.04 ng/mL. Therefore, ensuring adequate vitamin D levels could reduce the prevalence of IFG and T2DM, especially in females with high levels of testosterone.

## 1. Introduction

Diabetes is a chronic metabolic disease with high morbidity and mortality rates. A recent study has proposed that the estimated prevalence of prediabetes and diabetes was 50.1 and 11.6% in Chinese adults, respectively, which represented that there were 493.4 million people with prediabetes and 113.9 million people with diabetes in China [1]. Notably, the prevalence of diabetes has increased more rapidly in the rural population than the urban population [2]. Moreover, type 2 diabetes mellitus (T2DM) accounts for about 90% of the individuals with diabetes [3,4]. Impaired fasting glucose (IFG) is a state of glucose metabolism between diabetes and normal, and taking intervention at this stage can reduce the progression of diabetes effectively. Therefore, it is urgent to come up with effective strategies to prevent IFG and T2DM among the rural population in China.

Vitamin D, a fat-soluble vitamin, is essential for humans, including two forms: ergocalciferol (vitamin D2) and cholecalciferol (vitamin D3). Vitamin D deficiency has become a common health problem in China, especially in patients with IFG and T2DM [5]. Numerous studies have investigated the associations between vitamin D and IFG or T2DM, however, the results are controversial [6,7,8,9,10], and the underlying mechanisms are unclear. For instance, a prospective cohort study with 29 years of follow-up found that low 25(OH)D levels increased the risk of T2DM [9]. However, in a randomized, placebo-controlled trial, Pittas et al. found that a daily supplement of 4000 IU of vitamin D3 did not reduce the risk of developing diabetes compared to the placebo group [11]. Importantly, most of the studies were focused on developed countries and urban areas, and evidence for the relationship between vitamin D and diabetes in rural areas remains scant. Moreover, some clinical trials found that the supplementation of vitamin D could affect the levels of the lipid profile, including lowering the total cholesterol (TC) and low-density lipoprotein cholesterol (LDL-C), and increasing high-density lipoprotein cholesterol (HDL-C) levels [12,13,14]. A cross-sectional study found the positive associations between vitamin D and TC and triglyceride (TG) [15]. In addition, the lipid profile (TC, TG, HDL-C, and LDL-C) was associated with diabetes. However, whether the lipid profile mediated the effect of vitamin D on diabetes is still unclear.

Substantial basic and clinical evidence has indicated that vitamin D was related to serum testosterone, and the supplementation of vitamin D could increase serum testosterone levels [16,17,18]. Moreover, a cross-sectional study of metabolic diseases and risk factors among Chinese males in eastern China proposed that individuals who had lower levels of 25(OH)D were more likely to have lower levels of testosterone [19]. In addition, extensive cross-sectional studies have demonstrated an association between low testosterone and diabetes. Longitudinal studies also found that patients with diabetes had lower baseline testosterone levels compared to the control group [20]. Furthermore, several cohort studies such as the Framingham Heart study and the Western Australian Health in Men study yielded the consistent results that men with diabetes had lower testosterone levels than those without diabetes [21]. Moreover, our previous study found that serum testosterone levels were positively associated with T2DM in females [22]. However, whether serum vitamin D modified the effect of testosterone on IFG or T2DM remains largely unknown.

Therefore, we designed this study to explore the relationships of 25(OH)D_3_, 25(OH)D_2_, and total 25(OH)D with IFG or T2DM, and further ascertained whether the lipid profile mediated the relationships, and then to investigate the interaction of serum vitamin D and testosterone levels with IFG or T2DM by sex in Chinese rural adults.

## 2. Materials and Methods

### 2.1. Study Design and Population

The participant flow chart is shown in Appendix A. Participants in the present study were from an ongoing cohort study in rural areas of Henan Province (Registration Number: ChiCTR-OOC-15006699) [23]. In part I, we included 925 individuals with T2DM aged 18–79 years. According to the principle of the same sex and age within 3 years with T2DM patients, 925 IFG patients and 925 individuals with normal glucose tolerance (NGT) were selected. Participants lacking vitamin D information (*n* = 110), or for whom the concentration of vitamin D was extreme (*n* = 6) were excluded. Finally, 2659 participants were recruited, including 849 T2DM patients (324 males and 525 females), 913 IFG patients (352 males and 561 females), and 897 individuals with NGT (344 males and 553 females). In part II, since we conducted the principal component analysis (PCA) of blood lipid profile (TC, TG, HDL-C, and LDL-C) and mediation analysis to estimate whether the lipid profile (TC, TG, HDL-C, and LDL-C) mediated the association of vitamin D with IFG or T2DM, so we further excluded participants who are missing information on TC (*n* = 2), TG (*n* = 18), body mass index (BMI) (*n* = 8) or systemic blood pressure (SBP) (*n* = 3). Finally, 2628 participants were included in the analysis. The present study was performed in accordance with the Helsinki Declaration of 1975, and approved by the Life Science Ethic Committee of Zhengzhou University (Code: [2015] MEC (S128)). All participants provided their written informed consent prior to their inclusion in the study.

### 2.2. Data Collection

Structured questionnaires were used to obtain information on sociodemographic characteristics by well trained investigators, including age, sex, average monthly individual income, levels of education as well as family history of T2DM, and behavior habits, including smoking status (never smoking, previous smoker, and current smoker), alcohol intake (never drinking, previous drinker, and current drinker), and physical activity (low, mediate, and high). The grouping criteria for these variables have been described previously [22]. The participants were considered with a family history of T2DM if their parents or siblings had a history of T2DM. Anthropometric measurements were also collected in this study, such as weight, height, and blood pressure level. In brief, BMI was calculated by weight (kg) divided by square of height (m^2^). Blood pressure measurement was repeated three times with a sphygmomanometer (HEM-7071A), recording the SBP as well as diastolic blood pressure (DBP) by each measurement and taking its average value for analysis, and pulse pressure (PP) was calculated by SBP minus DBP. Using an updated homeostasis model (HOMA2) to estimate the *β* cell function and insulin resistance, and defined as HOMA2-β and HOMA2-IR [24].

### 2.3. Laboratory Measurements

Venous blood was drawn after overnight fasting (>8 h) and transferred immediately to a non-heparinizing tube for centrifugation after clotting, and then researcher placed the serum samples in a refrigerator at −80 °C. High performance liquid chromatography technology was used to determine glycosylated hemoglobin A1c (HbA1c). A radioimmunoassay was used to determine insulin (INS). Fasting plasma glucose (FPG) and lipid levels including TG, TC, LDL-C, as well as HDL-C were detected directly or enzymatically by using the Cobasc501 automatic biochemical analyzer.

Serum 25(OH)D_3_, 25(OH)D_2_, and testosterone levels were detected with a well-standardized liquid chromatography–tandem mass spectrometry system (LC–MS/MS) (a Waters XEVO TQ-S system (Waters, Milford, MA, USA)). The values below the limit of quantitation (LOQ) were replaced by half of the limit of detection (LOD). The value of total 25(OH)D was the sum of 25(OH)D_3_ and 25(OH)D_2_. Participants were equally divided into quartiles according to concentrations of serum 25(OH)D_3_, 25(OH)D_2_, and total 25(OH)D, and deemed the quartile 1 (Q1, the first/lowest quartile) as the reference group.

### 2.4. Ascertainment of Cases

According to the diagnostic criteria of the American Diabetes Association (ADA) (2002) and the WHO (1999), after excluding other types of diabetes, those who met one of the following conditions were diagnosed as IFG patients: (1) 6.1 mmol/L ≤ FPG < 7.0 mmol/L; (2) 5.7% ≤ HbA1c < 6.5%. Those who met one of the following conditions were diagnosed as T2DM patients: (1) FPG ≥ 7.0 mmol/L; (2) HbA1c ≥ 6.5%; (3) a self-reported history of T2DM and having taken anti-glycemic drugs in the past two weeks. Moreover, the individuals were grouped as underweight, normal, overweight, and obese based on BMI values, and the cut-points values were 18.5, 24.0, 28.0 kg/m^2^, respectively. According to the Chinese Guidelines for the Management of Hypertension (2005), a person who met one of the following criteria was diagnosed as hypertensive: (1) SBP ≥ 140 mmHg; (2) DBP ≥ 90 mmHg; (3) a self-reported history of hypertension and treating with anti-hypertensive drugs in the last two weeks. In addition, dyslipidemia was defined as TC ≥ 6.22 mmol/L, TG ≥ 2.26, LDL-C ≥ 4.14, HDL-C < 1.04 mmol/L, or taking lipid-lowering drugs in the past two weeks.

### 2.5. Statistical Analysis

Continuous data were represented by the median (interquartile ranges (IQR)) (skewed distribution) or mean (standard deviations (SD)) (normal distribution), while categorical data were presented by numbers (percentages). In addition, the differences in characteristics between cases and controls were assessed by Mann–Whitney U tests, Student’s t tests, and chi-square tests, respectively.

Since testosterone levels, HOMA2-*β*, and HOMA2-IR were skewed in terms of distribution, the natural logarithm was performed on them and then defined them as ln-testosterone, ln-HOMA2-*β*, and ln-HOMA2-IR before analysis. Logistic regression models were used to determine the odds ratios (*OR*s) and 95% confidence intervals (*CI*s) of presenting IFG or T2DM as a function of vitamin D levels. Three models were constructed. Model 1 was not adjusted. Model 2 was adjusted for smoking status, alcohol intake, physical activity, average monthly individual income, and level of education. Model 3 was further adjusted for model 2 + BMI, SBP, PP, TC, TG, HDL-C, LDL-C, and family history of T2DM. Trend tests were performed by inputting the categorical variables as continuous variables in logistic regression models to explore the relationship across increasing quartiles. In addition, linear regression analysis was performed to investigate the correlations of serum vitamin D levels with FPG, HbA1c, INS, ln-HOMA2-*β*, and ln-HOMA2-IR in model 3.

Furthermore, since age, sex, BMI [25], hypertension [26], and dyslipidemia [27] have effects on the chance of presenting IFG and T2DM, we further conducted sensitivity analyses to test the robustness of the results by considering age (<60 years or ≥60 years), sex (male or female), BMI (<24 kg/m^2^ (underweight/normal) or ≥24 kg/m^2^ (overweight/obesity)), hypertension (no or yes), and dyslipidemia (no or yes) in model 3 but not adjusting for BMI.

Principal component analysis (PCA) of the blood lipid profile (TC, TG, HDL-C, and LDL-C) was performed to create continuous variables representing individual lipid levels. The eigenvalue ≥ 1 was seen as a cutoff for factor retention. Mediation analysis was used to estimate whether the lipid profile mediated the association of vitamin D with IFG or T2DM and then assessed individually to ascertain which component of the lipid profile effected. The analysis was conducted in PROCESS procedure of SPSS with bootstrap 5000. In the meditation analysis, vitamin D was regarded as an independent variable, IFG or T2DM were regarded as the outcome, and the principal component I (PC I) and principal component II (PC II), which represent lipid profile, as well as TC, TG, HDL-C, and LDL-C, were separately used as mediating variables.

Generalized linear models and interactive plots were utilized to explore the interaction of serum vitamin D and testosterone with IFG or T2DM.

All analyses in this study were conducted with SPSS 21.0 and R 3.6.2. A two-sided *p* < 0.05 indicated statistically significant.

## 3. Results

### 3.1. Characteristics of the Participants

As Table 1 shows, individuals with IFG and T2DM were more likely to be with higher BMI, SBP, PP, TC, TG, FPG, HbA1c, and INS, as well as lower HDL-C, 25(OH)D_2_, and total 25(OH)D compared with those with NGT. Furthermore, the IFG individuals had a higher concentration of LDL-C and lower level of 25(OH)D_3_; T2DM individuals tended to have a family history of T2DM, to be smokers, and to have a lower level of physical activity than individuals with NGT. IFG and T2DM patients have lower levels of testosterone in males compared with NGT, and individuals with T2DM have higher testosterone levels in females compared to NGT.

### 3.2. Associations of Serum 25(OH)D_3_, 25(OH)D_2_ and Total 25(OH)D with Glucose Metabolism

The associations of serum 25(OH)D_3_, 25(OH)D_2_, and total 25(OH)D with glucose metabolism in quartiles and continuous variables are shown in Figure 1 and Appendix A. After adjusting the smoking status, alcohol intake, physical activity, average monthly individual income, level of education, BMI, SBP, PP, TC, TG, HDL-C, LDL-C, and family history of T2DM, per one unit increase in serum 25(OH)D_3_, 25(OH)D_2,_ or total 25(OH)D was related to a 1% (*OR* (95% *CI*): 0.99 (0.97, 1.00)), 15% (*OR* (95% *CI*): 0.85 (0.82, 0.88)), or 3% (*OR* (95% *CI*): 0.97 (0.96, 0.98)) lower chance of presenting IFG. Per one unit increase in serum 25(OH)D_2_ or total 25(OH)D was related to a 16% (*OR* (95% *CI*): (0.84 (0.81, 0.88)) or 3% (*OR* (95% *CI*): 0.97 (0.96, 0.99)) lower chance of presenting T2DM. Compared with the Q1, individuals with 25(OH)D_3_ in the Q4 was inversely related to IFG, and in the Q2 and Q3 were inversely related to T2DM (all *p* < 0.05). Individuals with 25(OH)D_2_ and total 25(OH)D in the Q3 or Q4 group were associated with a lower chance of presenting IFG and T2DM vs. those in the Q1 group (all *p* < 0.05). In addition, the risk of IFG and T2DM decreased with the levels of 25(OH)D_2_ and total 25(OH)D increased (all *p* for trend <0.05).

As shown in the Appendix A, serum 25(OH)D_3_ levels were negatively related to INS and ln-HOMA2-IR; 25(OH)D_2_ levels were negatively associated with FPG, HbA1c, INS, and ln-HOMA2-IR, and positively associated with ln-HOMA2-*β*; total 25(OH)D levels were inversely related to HbA1c, INS, and ln-HOMA2-IR.

### 3.3. Sensitivity Analyses

The results of sensitivity analyses were shown in Appendix A and Appendix A. The relationships between vitamin D and IFG or T2DM were not different according to age (< 60 years or ≥ 60 years), sex (male or female), BMI (<24 kg/m^2^ (underweight/normal) or ≥24 kg/m^2^ (overweight/obesity)), hypertension (no or yes), and dyslipidemia (no or yes).

### 3.4. Mediation Effects

As Appendix A showed, vitamin D was negatively associated with TC, TG, LDL-C, whereas it is positively associated with HDL-C. The PCA of the blood lipid profile (TC, TG, HDL-C, and LDL-C) was performed and two continuous variables (PC I and PC II*) were generated. PC I, characterized by TC and LDL-C; PC II, characterized by TG and HDL-C. The cumulative contribution rate accounts for 86.02% of the total variance:*PC I = 0.958738 × TC + 0.058608×TG + 0.368311×HDL-C + 0.937801 × LDL-C(1)
PC II = 0.244235 × TC + 0.912969×TG − 0.780856×HDL-C − 0.000072 × LDL-C(2)

Appendix A showed the schematic diagram of the mediation analysis. Figure 2 and Appendix A demonstrated the results of the mediation analysis. PC I or PC II was a potential modifier for the vitamin D-IFG and vitamin D-T2DM associations in males and females. In brief, PC I partly (2.74, 5.02 and 6.86%) mediated the associations between 25(OH)D_2_ and IFG in males and females, as well as the association between 25(OH)D and IFG in females; PC I mediated about 6.93% of the effect of 25(OH)D on T2DM in females. In addition, PC II mediated about 3.92% of the effect of 25(OH)D on T2DM in females; PC II mediated approximately 17.46 and 7.20% of the relationships between 25(OH)D and IFG in males as well as 25(OH)D_2_ and T2DM in females, respectively.

The results of the four lipid components (including TC, TG, HDL-C, and LDL-C) mediating the effects of vitamin D on IFG and T2DM were shown in Appendix A and Appendix A. TC partly mediated the relationship between 25(OH)D_2_ or total 25(OH)D and IFG or T2DM and the proportions explained were from 3.33 to 14.71%. TG mediated about 4.81 and 13.78% the effects of 25(OH)D on IFG in females and on T2DM in males, and 7.73% the effect of 25(OH)D_2_ on T2DM in females. LDL-C mediated approximately 2.36 and 4.38% of the relationships between 25(OH)D_2_ and IFG in males and females, respectively. In addition, HDL-C mediated 10.03% the effect of 25(OH)D on T2DM in males.

### 3.5. Interaction Effects

As shown in Table 2 and Figure 3, interactive effects of serum 25(OH)D_2_, total 25(OH)D, and testosterone on T2DM were observed in females (*p* for interactive = 0.002 and *p* for interactive = 0.028, respectively). Testosterone was positively correlated with T2DM, when total 25(OH)D was lower than 40.04 ng/mL or 25(OH)D_2_ was lower than 10.04 ng/mL. Nevertheless, testosterone was not correlated with T2DM, when total 25(OH)D was higher than 40.04 ng/mL. Testosterone was negatively correlated with T2DM, when 25(OH)D_2_ was higher than 21.81 ng/mL. In brief, serum 25(OH)D_2_ and total 25(OH)D counteracted the negative effect of testosterone on T2DM at certain levels among females.

## 4. Discussion

In the current study, we examined the associations between vitamin D and IFG or T2DM, then further explored whether the lipid profile mediated these associations. In addition, we investigated the interaction of testosterone and vitamin D with IFG or T2DM by sex. We found that higher levels of serum 25(OH)D_3_, 25(OH)D_2_, and total 25(OH)D were related to a lower chance of presenting IFG and T2DM, and a lipid profile mainly involving TC and TG could partially explain these relationships. Moreover, serum 25(OH)D_2_, total 25(OH)D counteracted the negative effect of testosterone on T2DM at certain levels among females.

Numerous previous studies have reported the association between vitamin D and blood glucose metabolism based on population, and some are consistent with our findings [9,28,29,30,31,32,33]. For instance, several cross-sectional studies proposed the modest negative correlation between total 25(OH)D and T2DM [31,32,33]. Additionally, a series of prospective studies also reported that low levels of serum total 25(OH)D increased the risk of IFG and T2DM [7,9,28,29]. In addition, Danting Li et al. advocated that elevated total serum 25(OH)D levels were related to a reduced risk of impaired glucose homeostasis among adults without diabetes in southwest China [6], which is supportive of our study. Most importantly, there is mounting evidence that vitamin D supplementation could reduce the chance of presenting IFG and T2DM by improving *β* cell function and insulin sensitivity and lowering fasting insulin, HbA1c, and insulin resistance [34,35,36,37,38]. Furthermore, a prospective case-cohort study based on the German arm of the European Prospective Investigation into Cancer and Nutrition (EPIC) study reported that, after adjusting for waist circumference and BMI, serum 25(OH)D_3_ was negatively related to T2DM among participants whose serum 25(OH)D_3_ concentrations were less than 45 nmol/L (22.08 ng/mL) [39], which is also consistent with our result. On the contrary, Ju-Sheng Zheng et al. found that the correlation between 25(OH)D_2_ and T2DM was not statistically significant [40], and some previous clinical trials demonstrated that supplementing vitamin D was not associated with oral glucose tolerance among people at risk for T2DM [8,11,41]. Moreover, other studies proposed that supplementing vitamin D has no effect on improving insulin sensitivity and secretion and glycemic control or slowing down the progression to diabetes in prediabetes [42], which are inconsistent with the results of this study. Possibly, these different results might be attributed to differences in ethnicity, economic levels, sample sizes, and vitamin D levels. Those studies were all conducted in developed counties with small sample sizes, and only including participants with vitamin D deficiency. However, the present study was conducted in rural China with a larger sample size, and including participants with all levels of vitamin D.

Previous studies reported that a low vitamin D status was associated with dyslipidemia, and the supplementation of vitamin D decreased TC and LDL-C levels in infertile women with polycystic ovary syndrome [13]. The same results were also found in vitamin D deficient HIV-infected patients [43]. In the Caerphilly Prospective Cohort Study, they found that higher vitamin D intake was associated with increased HDL-C [14]. A study conducted in Saudi women proposed positive correlations between serum vitamin D and TC and TG [15]. These results are consistent with the findings of this study that vitamin D was negatively associated with TC, TG, LDL-C, whereas they were positively associated with HDL-C levels. In addition, low vitamin D had negative effects on IR and glucose homeostasis [44]. As known, high lipid levels were the potential risk factors for diabetes. Moreover, our previous research found sex differences in lipid levels [45]. These results indicated that the lipid profile might mediate the role of vitamin D in diabetes status and differ in males and females. Thus, we conducted a mediation analysis and found that the lipid profile—mainly TC and TG—partly mediated the relationship between 25(OH)D_2_ or 25(OH)D and IFG or T2DM. This means that when vitamin D levels were high, the chance of developing IFG and T2DM could be decreased by reducing TC and TG levels. The possible mechanisms were that low vitamin D status induced secondary hyperparathyroidism, which caused calcium to spill out into fat cells and thus increased lipogenesis [46], and the increase in calcium ions in adipocytes increased the expression of fatty acid synthase, which was a key regulator of lipid deposition and reduced lipolysis [47]. Moreover, vitamin D stimulated the mRNA expression and secretion of leptin [48], and leptin control lipid metabolism through the inhibition of lipogenesis and stimulation of lipolysis [49]. This leads to high lipid deposition in the liver, which interferes with insulin signaling and eventually leads to *β* cell dysfunction [50]. The underlying mechanism by which vitamin D affects diabetes status through altering the lipid profile is not yet clear, and a more rigorous study is needed.

Up to date, several studies have investigated the association of vitamin D and testosterone. The positive relationship between serum total 25(OH)D and testosterone levels has been reported in males. The same result was also observed in animal trials, suggesting that the serum testosterone levels in vitamin D deficient rats could increase to normal after vitamin D supplementation [51]. At the population level, a cross-sectional study which collected data from 16 sites in East China proposed that lower total 25(OH)D levels were significantly correlated with lower levels of testosterone in Chinese males [19]. A randomized controlled trial in which participants received either 83 μg (3332 IU) vitamin D (*n* = 31) or placebo (*n* = 23) daily for 1 year, observed that supplementing with vitamin D could raise the testosterone levels in males [52]. However, few studies reported the association of vitamin D with serum testosterone levels among females, and only some clinical studies in individuals with polycystic ovary syndrome proposed that the experimental group treated with vitamin D supplementation had lower serum testosterone levels and a lower incidence of insulin resistance compared with the control group. Those indicated that the effect of vitamin D on the serum testosterone levels may be opposed between males and females. Moreover, several observational studies and randomized trials have proposed that low serum testosterone status were correlated with an increased risk of IFG and T2DM in males and the opposite in females [22,53,54,55,56]. It seems plausible that males with higher vitamin D levels likely have higher testosterone levels, which decrease the risk of IFG and T2DM. On the other hand, females with higher vitamin D levels are likely to have lower testosterone levels, which decreases the risk of IFG and T2DM.

The biological mechanism of the interaction between vitamin D and testosterone are not well understood, and there are several possible interpretations. On the one hand, vitamin D metabolizing enzymes and vitamin D receptor (VDR) are expressed in the human reproductive system, especially in testicular and ovarian tissues. In males, vitamin D binds to the VDR and upregulates certain testis-specific genes such as ATP-binding cassette transporter A1 (ABCA1) to stimulate testosterone secretion [57]. In females, vitamin D combines with the VDR in ovaries [58], and then combines with an atypical vitamin D response element that is located in cytochrome P450 family 19 subfamily A member 1 (CYP19A1) promoter to encode P450 aromatase, and P450 aromatase catalyzes the conversion of androgen to estrogen, and results in a decrease in serum testosterone levels. In addition, testosterone inhibits the expression of protein tyrosine phosphatase 1B (PTP1B) which mediated by hypothalamic NF-κ B to regulate insulin sensitivity [59], meanwhile, testosterone may reduce the concentration of adiponectin (insulin-sensitive lipid-cell derived protein) in plasma, thus improving the insulin sensitivity by combining with androgen receptors [60]. On the other hand, androgen increases 1α-hydroxylase, a key enzyme in vitamin D metabolism, turning 25(OH)D into 1, 25(OH)_2_D, a more active form, and 1, 25(OH)_2_D reduces the risk of IFG and T2DM by promoting insulin synthesis and secretion, increasing insulin sensitivity, and reducing insulin resistance. The underlying mechanism of this interaction is complex and needs to be further investigated.

In the current study, we conducted an interaction analysis to explore whether the effects of serum vitamin D and testosterone levels on IFG or T2DM in Chinese rural population. We found that serum 25(OH)D_2_ and total 25(OH)D counteracted the negative effect of testosterone on T2DM at certain levels in females.

There are several limitations in the current study. First of all, this study was a case-control design, hence the causal relationship cannot be drawn. Secondly, the population of the study came from the rural area of Henan province, so the conclusions may not apply to urban populations. Thirdly, although we considered and adjusted many confounding factors, the possibility of other potential confounders cannot be ruled out.

## 5. Conclusions

Serum 25(OH)D_3_, 25(OH)D_2_, and total 25(OH)D levels were inversely associated with IFG and T2DM in the Chinese rural population. The levels of lipid profile including TC, TG, and LDL-C were positively associated with the chance of presenting IFG and T2DM, whereas HDL-C was negatively associated with the chance of presenting IFG and T2DM. Moreover, the lipid profile—mainly TC and TG—partly mediated the relationship between 25(OH)D_2_ or 25(OH)D and IFG or T2DM. In addition, the interaction between serum 25(OH)D_2_, total 25(OH)D, and testosterone on T2DM were observed in females, and serum 25(OH)D_2_, total 25(OH)D counteracted the positive association between testosterone and T2DM at certain levels. Therefore, ensuring adequate vitamin D levels could reduce the risk of IFG and T2DM by reducing the lipid profile. Combined with the findings of this study and previous studies, it can be known that people with a low vitamin D level and females with higher testosterone levels may have a higher chance of presenting glucose metabolism disorder, which provides a scientific basis for the clinical screening of high-risk groups of diabetes. More importantly, this study laid a foundation for the research of the mechanism of vitamin D, lipid profile and testosterone on diabetes, and proposed new strategies for the prevention of diabetes.

## Figures and Tables

**Figure 1 nutrients-13-00090-f001:**
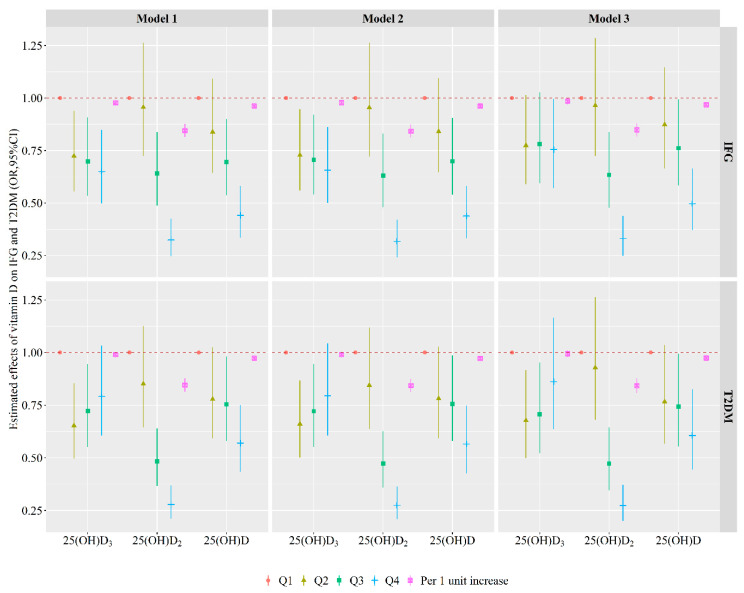
Associations of serum 25(OH)D_3_, 25(OH)D_2_, and total 25(OH)D levels with IFG and T2DM. Model 1: no adjust; Model 2: adjusted for smoking status, alcohol intake, physical activity, average monthly individual income, level of education; Model 3: model 2 + BMI, SBP, PP, TC, TG, HDL-C, LDL-C and family history of T2DM. Abbreviations: IFG, impaired fasting glucose; *OR*, odds ratio; Q1, the first/lowest quartile; Q2, the second quartile; Q3, the third quartile; Q4, the fourth/highest quartile; T2DM, type 2 diabetes mellitus; 95% *CI*: 95% confidence interval.

**Figure 2 nutrients-13-00090-f002:**
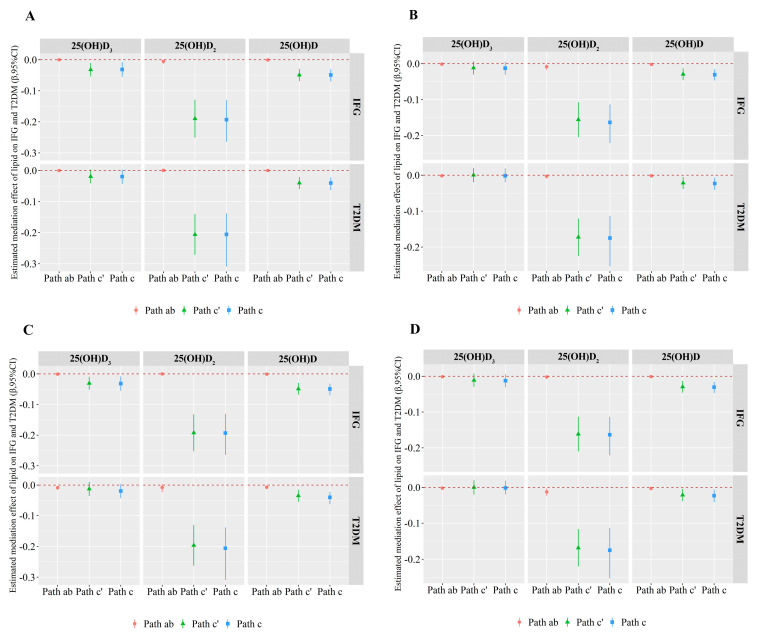
Mediation analysis of the relationships between 25(OH)D_3_, 25(OH)D_2_ or 25(OH)D and IFG or T2DM by lipid levels in males and females, respectively. Path ab, Path c’ and Path c represent the indirect, direct and total effects of vitamin D on diabetes status with lipid levels (PC I and PC II) as a mediator in males and females, respectively. (**A**) displays the mediation effect of PC I on the association between 25(OH)D_3_, 25(OH)D_2_ or 25(OH)D and IFG or T2DM in males; (**B**) displays the mediation effect of PC I on the association between 25(OH)D_3_, 25(OH)D_2_ or 25(OH)D and IFG or T2DM in females; (**C**) displays the mediation effect of PC II on the association between 25(OH)D_3_, 25(OH)D_2_ or 25(OH)D and IFG or T2DM in males; (**D**) displays the mediation effect of PC II on the association between 25(OH)D_3_, 25(OH)D_2_ or 25(OH)D and IFG or T2DM in females. Adjusted for alcohol intake, smoking status, physical activity, average monthly individual income, level of education, BMI, SBP, PP, and a family history of T2DM. Abbreviations: IFG, impaired fasting glucose; T2DM, type 2 diabetes mellitus; 95% *CI*: 95% confidence interval.

**Figure 3 nutrients-13-00090-f003:**
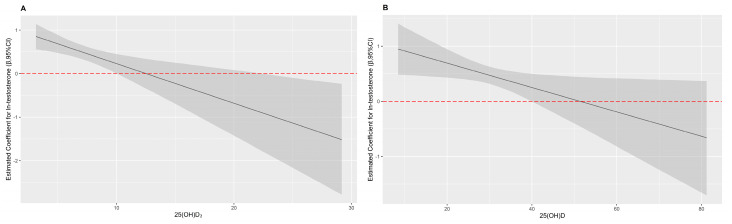
The interaction plots of 25(OH)D_2_, total 25(OH)D and testosterone on T2DM in females. (**A**) the effect of ln-testosterone on T2DM at different levels of 25(OH)D_2_; (**B**) the effect of ln-testosterone on T2DM at different levels of total 25(OH)D; adjusted for smoking status, alcohol intake, physical activity, average monthly individual income, level of education, BMI, SBP, PP, TC, TG, HDL-C, LDL-C and a family history of T2DM.

**Table 1 nutrients-13-00090-t001:** Characteristics of the participants according to diabetes status.

Variables	NGT	IFG	T2DM	*p* Value	
				IFG vs. NGT	T2DM vs. NGT
Subjects, *n*	897	913	849		
Age, (years)	61 (54, 66)	61 (54, 66)	61 (54, 66)	0.961	0.970
Male, *n* (%)	344 (38.35)	352 (38.55)	324 (38.16)	0.929	0.936
BMI, (kg/m^2^)	23.42 (21.18, 25.75)	24.22 (22.12, 26.53)	25.43 (23.48, 27.68)	<0.001	<0.001
Smoking status, *n* (%)				0.700	0.038
Never smoking	667 (74.36)	692 (75.79)	625 (73.62)		
Previous smoker	57 (6.35)	59 (6.46)	80 (9.42)		
Current smoker	173 (19.29)	162 (17.74)	144 (16.96)		
Alcohol intake, *n* (%)				0.506	0.298
Never drinking	737 (82.16)	766 (83.90)	675 (79.51)		
Previous drinker	48 (5.35)	49 (5.37)	58 (6.83)		
Current drinker	112 (12.49)	98 (10.73)	116 (13.66)		
Physical activity, *n* (%)				0.520	0.040
Low	213 (23.75)	238 (26.07)	245 (28.86)		
Mediate	453 (50.50)	446 (48.85)	388 (45.70)		
High	231 (25.75)	229 (25.08)	216 (25.44)		
Average monthly individual income, *n* (%)				0.228	0.309
<500, CNY	352 (39.24)	381 (41.73)	345 (40.64)		
500~, CNY	279 (31.10)	294 (32.20)	236 (27.80)		
1000~, CNY	266 (29.65)	238 (26.07)	268 (31.57)		
Level of education, *n* (%)				0.749	0.515
Never attended school	238 (26.53)	228 (24.97)	207 (24.38)		
Primary school	271 (30.21)	281 (30.78)	255 (30.03)		
Junior secondary and above	388 (43.26)	404 (44.25)	387 (45.58)		
Family history of T2DM, *n* (%)	15 (1.67)	14 (1.53)	45 (5.30)	0.814	<0.001
SBP, (mmHg)	118.00 (108.00, 131.00)	121.00 (110.00, 134.00)	125.00 (114.00, 138.00)	<0.001	<0.001
PP, (mmHg)	45.00 (39.00, 53.00)	49.00 (42.00, 57.00)	47.00 (39.00, 56.00)	0.022	<0.001
TC, (mmol/L)	4.60 (0.80)	4.83 (4.25, 5.44)	4.79, (4.21, 5.50)	<0.001	<0.001
TG, (mmol/L)	1.41 (1.00, 1.95)	1.64 (1.11, 2.31)	1.92 (1.34, 2.94)	<0.001	<0.001
HDL-C, (mmol/L)	1.37 (1.16, 1.62)	1.32 (1.12, 1.57)	1.23 (1.02, 1.47)	0.010	<0.001
LDL-C, (mmol/L)	2.82 (0.71)	3.00 (2.47, 3.50)	2.83 (0.95)	<0.001	0.583
FPG, (mmol/L)	4.94 (0.46)	5.30 (4.89, 5.82)	7.94 (7.00, 10.21)	<0.001	<0.001
HbA1c, (%)	5.30 (5.10, 5.50)	5.90 (5.70, 6.00)	7.40 (6.60, 8.90)	<0.001	<0.001
INS, (uIU/mL)	11.94 (9.31, 15.20)	12.59 (9.86, 16.33)	14.12 (11.40, 18.52)	0.001	<0.001
Testosterone, (ng/mL)					
Male	2.30 (1.70, 3.00)	2.00 (1.40, 2.70)	1.80 (1.30, 2.50)	<0.001	<0.001
Female	0.10 (0.02, 0.10)	0.10 (0.02, 0.10)	0.10 (0.10, 0.10)	0.053	<0.001
25(OH)D_3_, (ng/mL)	23.07 (19.02, 28.00)	22.07 (18.09, 27.03)	22.87 (18.30, 28.04)	<0.001	0.182
25(OH)D_2_, (ng/mL)	7.60 (6.04, 9.84)	6.46 (5.43, 7.85)	6.30 (5.23, 7.80)	<0.001	<0.001
25(OH)D, (ng/mL)	31.12 (26.29, 37.50)	29.26 (24.61, 34.23)	29.77 (24.84, 35.51)	<0.001	<0.001

Values are the mean (standard deviation) or median (inter-quartile range) for continuous variables and the number (percentages) for the categorical variable. *p* values were calculated using the *t*-test or Mann–Whitney U test and chi-square test. Abbreviations: BMI, body mass index; FPG, fasting plasm glucose; HbA1c, glycosylated hemoglobin; HDL-C, high-density lipoprotein cholesterol; IFG, impaired fasting glucose; INS, insulin; LDL-C, low-density lipoprotein cholesterol; NGT, normal glucose tolerance; PP, pulse pressure; CNY, renminbi; SBP, systolic blood pressure; TC, triglycerides; TG, triglyceride; T2DM, type 2 diabetes mellitus.

**Table 2 nutrients-13-00090-t002:** The multiplication interactive effects of 25(OH)D_3_, 25(OH)D_2_, or total 25(OH)D and testosterone on IFG and T2DM.

Variables	Male *OR*s (95% *CI*s)			Female *OR*s (95% *CI*s)		
	Vitamin D	Ln-Testosterone	*p* for Interaction	Vitamin D	Ln-Testosterone	*p* for Interaction
IFG						
25(OH)D_3_	0.97 (0.95, 0.99) *	-	-	0.99 (0.98, 1.01)	-	-
25(OH)D_2_	0.83 (0.78, 0.88) *	-	-	0.86 (0.82, 0.90) *	-	-
25(OH)D	0.95 (0.93, 0.97) *	-	-	0.98 (0.96, 0.99) *	-	-
Ln-testosterone	-	0.82 (0.63, 1.10)	-	-	1.12 (0.99, 1.28)	-
25(OH)D_3_ + Ln-testosterone	0.97 (0.95, 0.99) *	0.87 (0.66, 1.13)	-	1.00 (0.98, 1.01)	1.12 (0.99, 1.28)	-
25(OH)D_2_ + Ln-testosterone	0.83 (0.78, 0.88) *	0.90 (0.69, 1.17)	-	0.86 (0.81, 0.90) *	1.15 (1.01, 1.31) *	-
25(OH)D + Ln-testosterone	0.95 (0.94, 0.97) *	0.92 (0.70, 1.20)	-	0.98 (0.96, 0.99) *	1.12 (0.98, 1.28)	-
25(OH)D_3_ + Ln-testosterone + 25(OH)D_3_ × Ln-testosterone	0.98 (0.95, 1.02)	1.10 (0.47, 2.58)	0.559	0.95 (0.90, 1.01)	1.57 (1.02, 2.42)	0.109
25(OH)D_2_ + Ln-testosterone + 25(OH)D_2_ × Ln-testosterone	0.82 (0.74, 0.90) *	0.77 (0.29, 2.01)	0.739	0.91 (0.79, 1.06)	0.97 (0.66, 1.44)	0.370
25(OH)D + Ln-testosterone + 25(OH)D × Ln-testosterone	0.96 (0.92, 0.99) *	0.98 (0.33, 2.95)	0.906	0.95 (0.90, 1.00)*	1.51 (0.91, 2.50)	0.224
**T2DM**						
25(OH)D_3_	0.98 (0.96, 1.01)	-	-	1.00 (0.98, 1.02)	-	-
25(OH)D_2_	0.82 (0.77, 0.88) *	-	-	0.86 (0.81, 0.90) *	-	-
25(OH)D	0.96 (0.94, 0.98) *	-	-	0.98 (0.97, 1.00) *	-	-
Ln-testosterone	-	0.76 (0.58, 1.00)	-	-	1.61 (1.38, 1.87) *	-
25(OH)D_3_ + Ln-testosterone	0.99 (0.96, 1.01)	0.78 (0.59, 1.03)	-	1.01 (0.99, 1.03)	1.61 (1.38, 1.88) *	-
25(OH)D_2_ + Ln-testosterone	0.83 (0.77, 0.88) *	0.82 (0.62, 1.09)	-	0.85 (0.81, 0.90) *	1.62 (1.38, 1.89) *	-
25(OH)D + Ln-testosterone	0.97 (0.95, 0.99) *	0.82 (0.62, 1.09)	-	0.99 (0.97, 1.00)	1.60 (1.37, 1.86) *	-
25(OH)D_3_ + Ln-testosterone + 25(OH)D_3_ × Ln-testosterone	0.99 (0.95, 1.02)	0.85 (0.37, 1.98)	0.838	0.96 (0.90, 1.02)	2.51 (1.45, 4.37) *	0.099
25(OH)D_2_ + Ln-testosterone + 25(OH)D_2_ × Ln-testosterone	0.81 (0.73, 0.90) *	0.64 (0.25, 1.65)	0.582	0.66 (0.55, 0.79) *	3.14 (1.99, 4.96) *	0.002
25(OH)D + Ln-testosterone + 25(OH)D × Ln-testosterone	0.96 (0.93, 0.99) *	0.68 (0.24, 1.95)	1.006	0.93 (0.88, 0.98) *	3.14 (1.68, 5.88) *	0.028

Adjusted for smoking status, alcohol intake, physical activity, average monthly individual income, level of education, BMI, SBP, PP, TC, TG, HDL-C, LDL-C and a family history of T2DM. * *p* < 0.05. Abbreviations: IFG, impaired fasting glucose; *OR*, odds ratio; T2DM, type 2 diabetes mellitus; 95%*CI*: 95% confidence interval.

## Data Availability

No new data were created or analyzed in this study. Data sharing is not applicable to this article.

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
