# Peer review of "Serum Vitamin D Affected Type 2 Diabetes though Altering Lipid Profile and Modified the Effects of Testosterone on Diabetes Status"

_nutrients, 2020, doi:10.3390/nu13010090_

Round 1
Reviewer 1 Report
In the current study, Wang et al investigated if the effects of serum vitamin D and testosterone levels on IFG or T2DM in the Chinese rural population and reported that serum 25(OH)D2, total 25(OH)D counteracted the negative effect of testosterone on T2DM at certain levels in females.
The study is important and provides some interesting insights regarding serum testosterone and Vitamin D interplay affecting glucose metabolism, however, some points need to be addressed-
1- In the abstract at least one sentence needs to be added regarding background emphasizing why this study is needed.
2- Abbreviations like IGF and T2DM must be elaborated in the abstract and subsequently ai the first instance.
3- The introduction is too short. There must be details regarding the role of testosterone on diabetes. Needs some more references and mention of previous studies highlighting the gaps.
4- The clinical significance of the finding needs to be discussed.
Reviewer 2 Report
Dear Authors,
Thank you for the opportunity to review your paper entitled Serum vitamin D was related to diabetes status and 2 modified the effects of testosterone on diabetes status.
As a former clinician and current translational medicine researcher, I appreciate all the clinical studies with large cohorts elucidating important metabolic paper.
This time, first of all, I need some clarifications and comments regarding your paper.
Major issues:
- The main point of the study from my point of view is to reveal the existence of interactions between T2DM, it D, and testosterone - it gives a freshness to the paper. Your main limitation is the fact that you took into account mainly testosterone levels in females. It is fine but did you consider the fluctuations of the levels of testosterone during the female period cycle? Did you consider daily fluctuations of its level during the span of the day or day/night cycle?
- Moreover, the testosterone levels are very low in all the females. Taking into account that you use a very precise method of the assays, I am worried that your levels are just artificial. It might explain weird results: 0.1 (0.02, 0.1) and for T2DM 0.1 (0.1, 0.1). From my experience, it means that the levels very barely undetectable and the precision of the assay went dramatically down. I would not trust these results.
- I am also very concerned about the design of the experiment. You have patients with the age ranging from 18 to 79. This is an incredible wide span. Accordingly to American guidelines you should divide them into subgroups to eliminate the bias due to this matter. Moreover, you selected randomly IFG & NGT subjects. I do not think that randomly selected subjects might have the perfect distribution of sex percentage, age, and others.
- I do not understand the way of presenting data. Normal distribution should be presented as means±SD/SEM and skewed distribution as median (full range/75%quart).
- Moreover, the rest of the results are barely non-significant. Table 2 shows lots of insignificant data that is weird taking into account really fair number of patients.
- The conclusions are not supported by the results. Furthermore, these results can not be extrapolated to the general population. You focused on a very narrow rural population. In the discussion, there is a lack of any impact of genetic/phenotype variations on the biases in the results. Once again, your conclusions are hypotheses - they are not supported by the results.
Minors:
- Please do not use abbreviations without their explanation in the abstract - IFG stands for interferon-gamma.
- The quality of the Figures is very low. Must be improved.
- English used in your manuscript is not fair enough for American English users. The constructions, grammar inequalities, and weird order of the words make the paper really hard-to-read.
Round 2
Reviewer 2 Report
The authors correctly responded to all my questions and concerns.
Author Response
Thank you very much for your constructive comment and suggestions.
